**Data Availability Statement:** No datasets were generated or analyzed for the current study

# OXIDATIVE study: A pilot prospective observational cohort study protocol examining the influence of peri-reperfusion hyperoxemia and immune dysregulation on early allograft dysfunction after orthotopic liver transplantation

Elizabeth A. Wilson [1]*, Anna Woodbury [1], Kirsten M. Williams[2], Craig M. Coopersmith[3]

**1** Department of Anesthesiology, Emory University School of Medicine, Emory University Hospital, Atlanta, GA, United States of America, **2** Department of Pediatrics, Division of Hematology and Oncology, Emory University School of Medicine, Children's Hospital of Atlanta, Atlanta, GA, United States of America, **3** Department of Surgery and Emory Critical Care Center, Emory University School of Medicine, Emory University Hospital, Atlanta, GA, United States of America

* ewilso5@gmail.com

## Abstract

Early allograft dysfunction (EAD) is a functional hepatic insufficiency within a week of orthotopic liver transplantation (OLT) and is associated with morbidity and mortality. The etiology of EAD is multifactorial and largely driven by ischemia reperfusion injury (IRI), a phenomenon characterized by oxygen scarcity followed by paradoxical oxidative stress and inflammation. With the expanded use of marginal allografts more susceptible to IRI, the incidence of EAD may be increasing. This necessitates an in-depth understanding of the innate molecular mechanisms underlying EAD and interventions to mitigate its impact. Our central hypothesis is peri-reperfusion hyperoxemia and immune dysregulation exacerbate IRI and increase the risk of EAD. We will perform a pilot prospective single-center observational cohort study of 40 patients. The aims are to determine (1) the association between peri-reperfusion hyperoxemia and EAD and (2) whether peri-reperfusion perturbed cytokine, protein, and hypoxia inducible factor-1 alpha (HIF-1α) levels correlate with EAD after OLT. Inclusion criteria include age $\geq$ 18 years, liver failure, and donation after brain or circulatory death. Exclusion criteria include living donor donation, repeat OLT within a week of transplantation, multiple organ transplantation, and pregnancy. Partial pressure of arterial oxygen ($PaO_2$) as the study measure allows for the examination of oxygen exposure within the confines of existing variability in anesthesiologist-administered fraction of inspired oxygen ($FiO_2$) and the inclusion of patients with intrapulmonary shunting. The Olthoff et al. definition of EAD is the primary outcome. Secondary outcomes include postoperative acute kidney injury, pulmonary and biliary complications, surgical wound dehiscence and infection, and mortality. The goal of this study protocol is to identify EAD contributors that could be targeted to attenuate its impact and improve OLT outcomes. If validated, peri-reperfusion

protocol manuscript. All relevant data from this study will be made available upon study completion.

**Funding:** This study is supported by the Robert W. Woodruff Health Science Center and the National Center for Advancing Translational Sciences of the National Institutes of Health under Award Number UL1TR002378 (https://georgiactsa.org/funding/pilot-grants.html). The content is solely the responsibility of the authors and does not necessarily represent the official views of the Robert W. Woodruff Health Science Center or the National Institutes of Health. This study is also supported by the International Liver Transplantation Society (ILTS) Vanguard Research Award sponsored by Natera (https://ilts.org/about/awards/ilts-vanguard-research-grant-sponsored-by-natera/). The content is solely the responsibility of the authors and does not necessarily represent the official views of ILTS or Natera. EW is the recipient of both grant awards. The funders did not and will not have a role in study design, data collection and analysis, decision to publish, or preparation of the manuscript.

**Competing interests:** The authors have declared that no competing interests exist.

**Abbreviations:** ABG, Arterial blood gas; AKI, Acute kidney injury; ANOVA, Analysis of variance; ATP, Adenosine triphosphate; DCD, Donation after circulatory death; DAMPS, Damage-associated molecular patterns; EAD, Early allograft dysfunction; ERHCCL, Emory Research Hemostasis & Coagulation Core Lab; FIH, Factor inhibiting HIF; $FiO_2$, Fraction of inspired oxygen; GA CTSA, Georgia Clinical & Translational Science Alliance; GCRC, Georgia Clinical & Translational Science Alliance Research Center; HIF, Hypoxia inducible factor; HIF-1α, Hypoxia inducible factor 1 alpha; HO-1, Heme oxygenase 1; IFNγ, Interferon-gamma; IL, Interleukin; ILTS, International Liver Transplantation Society; IP-10, Interferon gamma-induced protein 10; IRB, Institutional review board; IRI, Ischemia reperfusion injury; IQR, Interquartile range; KDIGO, Kidney Disease Improving Global Outcomes; MASH, Metabolic dysfunction-associated steatohepatitis; MCP-1, Monocyte chemoattractant protein 1; MELD, Model for End-stage Liver Disease; MIG, Monokine induced by gamma interferon; NF-κB, Nuclear factor kappa-light-chain enhancer of activated B cell system; NIH, National Institutes of Health; NRP, Normothermic regional perfusion; OLT, Orthotopic liver transplantation; $PaO_2$, Partial pressure of arterial oxygen; PHD, Prolyl hydroxylase domain; PI, Principal investigator; PTX3, Pentraxin-related protein 3; RANTES, Regulated on activation,

hyperoxemia and immune perturbations could be targeted via $FiO_2$ titration to a goal $PaO_2$ and/or administration of an immunomodulatory agent by the anesthesiologist intraoperatively.

# Introduction

## Early allograft dysfunction (EAD) impact and pathophysiology

Early allograft dysfunction (EAD) is a functional hepatic insufficiency within one week of orthotopic liver transplantation (OLT). Known risk factors for EAD are shown in "Table 1" [1].

EAD occurs in 20–25% of deceased donor OLT recipients and is associated with increased hospital costs, morbidity, and mortality [2–10]. The etiology of EAD is multifactorial, including donor and recipient factors, and largely driven by ischemia reperfusion injury (IRI). IRI is a process characterized by dysregulation of cellular oxygen homeostasis and innate immune defenses in the allograft after temporary cessation (ischemia) and later restoration (reperfusion) of oxygen-rich blood flow "Fig 1" [11, 12].

During the ischemia phase, oxygen deprivation causes reduced energy accessibility, inflammation, and hepatocellular swelling [10–14]. Decreased adenosine triphosphate (ATP) production leads to acidosis, cellular membrane ion pump dysfunction, excess intracellular calcium, mitochondrial fragility, sinusoidal constriction, and hepatocellular damage [11–14]. During the reperfusion phase, the reintroduction of oxygen elicits a self-perpetuating cascade of reactive oxygen species (ROS) and inflammatory cytokine production that leads to hepatocellular congestion, thrombus formation, and apoptosis [11–14].

The reperfusion phase occurs in two phases: (1) the initial phase, which occurs within the first 2 hours, and (2) the late phase, which occurs 6–48 hours after reperfusion [11, 12]. The initial phase of reperfusion is characterized by the release of ROS and proinflammatory cytokines [11–13, 15, 16] that upregulate adhesion molecules on the surface of hepatocytes and sinusoidal endothelial cells (SEC) [11, 12, 15]. In the late phase of reperfusion, neutrophils and T-cells bind to adhesion molecules on hepatocytes and SEC and infiltrate the liver parenchyma [1–12, 15, 17], leading to mitochondrial permeability and the release of damage-associated molecular patterns (DAMPS) [11, 12]. Through the release of DAMPS and activation of transcription factors, such as nuclear factor kappa-light-chain enhancer of activated B cell system (NF-κB), injured hepatocytes continue this self-perpetuating cycle of ROS and inflammatory cytokine production that ultimately leads to dysregulated microcirculatory blood flow, apoptosis, and coagulative necrosis [11, 12, 15–20].

**Table 1. EAD risk factors by donor, procurement, and recipient characteristics.**

| Donor | Procurement | Recipient |
|---|---|---|
| Donor age > 45 | Donors after circulatory death | Age |
| Steatosis > 30% | Prolonged ischemia time | MELD Score |
| | | UNOS Status |

[1] Model for End-stage Liver Disease (MELD) score measures the severity of liver disease. United Network for Organ Sharing (UNOS) status is a numerical value used to assign relative priority for distributing donated organs.

normal T cell expressed and secreted; ROS, Reactive oxygen species; SD, Standard deviation; SEC, Sinusoidal endothelial cells; SOC, Standard of care; TNFα, Tumor necrosis factor alpha; UNOS, United Network for Organ Sharing.

## IRI as a critical barrier

To accommodate rising global demand for OLT in sicker patients and minimize death on the waitlist, the deceased donor pool has expanded with the use of extended criteria donors and marginal allografts more susceptible to IRI [21], including those with advanced age, donation after circulatory death (DCD), prolonged ischemia time, and steatosis. Hepatic IRI is a critical barrier to the current use and further expansion of the available deceased donor pool as the incidence of EAD and other poor postoperative outcomes may be increasing [22]. To mitigate the impact of IRI and EAD, normothermic regional perfusion (NRP) and hypothermic and normothermic machine perfusion have been implemented to restore allograft oxygenation pre-implantation. This new technology, however, has limited global accessibility and thus static cold storage of the allograft on ice remains the conventional and most widely used method of allograft storage worldwide.

## Effect of hyperoxemia on EAD

In a mouse model, postoperative hyperoxia with a fraction of inspired oxygen (FiO$_2$) of 60% (equating to an average partial pressure of arterial oxygen or PaO$_2$ of 230 mmHg) significantly increased IRI- induced hepatocellular damage compared to normoxia with a FiO$_2$ of 21% [23]. In a human living donor liver transplantation (LDLT) study, intraoperative hyperoxia with a

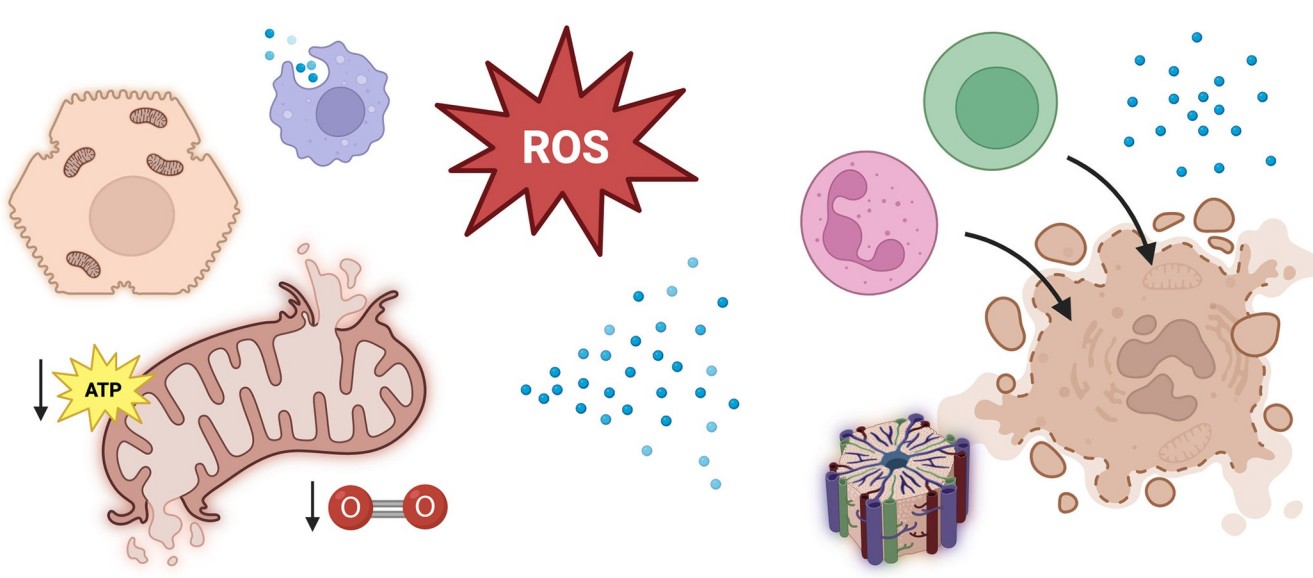

**Fig 1. Pathophysiology of hepatic ischemia reperfusion injury.** ATP indicates adenosine triphosphate; ROS, reactive oxygen species; DAMPS, damage associated molecular patterns; NF-κB, nuclear factor-kappaB; MMPs, matrix metalloproteinases; NO, nitric oxide. Figure created with Biorender.com.

calculated average $FiO_2 \geq 50\%$ was associated with EAD and worse allograft survival compared to $FiO_2 < 50\%$ [24].

There is no standard of care concerning the optimal peri-reperfusion oxygen level during OLT. Furthermore, the impact of peri-reperfusion excess oxygen exposure in deceased donor OLT recipients has not been examined. A 2022 expert panel of liver transplant clinicians recommended a restrictive oxygen strategy targeting a $PaO_2$ of 70–120 mmHg during OLT, citing evidence from non-OLT surgeries such as abdominal surgery [25]. These abdominal surgery studies demonstrate an increased risk of mortality, cancer, atelectasis, and elevated cardiac injury markers with higher perioperative oxygen concentrations [25–29]. However, without evidence in deceased donor OLT, this recommendation has not been widely implemented. Many OLT anesthesiologists administer a high peri-reperfusion $FiO_2$ to mitigate cellular hypoxia during malperfusion, while others administer a low $FiO_2$ to attenuate IRI. We hypothesize peri-reperfusion hyperoxemia contributes to IRI and increases the risk of EAD. Examining this is important because variations in clinical practice may contribute to IRI and worse postoperative outcomes if excess oxygen exposure is associated with EAD.

### Novel serum biomarkers for EAD

Multiple studies demonstrate a distinct cytokine profile associated with IRI in OLT [11, 30, 31]. In a study measuring 38 cytokines in 53 OLT recipients, data suggest donors and recipients differentially contribute to cytokine expression in biopsy-proven IRI [30]. In this study, preoperative peripheral blood from recipients with IRI showed increased expression of tumor necrosis factor alpha (TNF-α), interleukin (IL)-5 (IL-5), IL-13, IL-2, IL-7, IL-8, and IL-1Ra [30]. Early in the postoperative period, patients who developed IRI demonstrated increased peripheral expression of IL-1Ra, IL-4, IL-13, and IL-17A [30]. Peripheral and portal IL-8 levels remained significantly elevated at all intraoperative time points tested [30] and could serve as a target for intervention [32, 33].

Furthermore, perioperative cytokine aberrations linked to IRI have been studied later in the development of EAD. One exploratory nested case-control study examined the pattern of expression of 25 serum cytokine levels in 73 deceased donor OLT recipients, 29 of whom developed EAD [34]. Serum samples were obtained preoperatively on the day of transplant and over 90 days postoperatively. The development of EAD was associated with lower IL-6 and higher IL-2R levels preoperatively [34]. The odds of developing EAD based on preoperative IL-6 and IL-2R levels in the multivariate model were 0.20 (95% CI, 0.06–0.64, p = 0.006) and 3.77 (95% CI, 1.22–11.64, p = 0.021) respectively [34], demonstrating the pre-transplant milieu may contribute to IRI and EAD. The development of EAD was also associated with higher MCP-1, IL-8, RANTES, MIG, IP-10, and IL-2R levels postoperatively, suggesting upregulation of the NF-κB pathway and T cell activation in the early postoperative period [34].

Though studies have associated peri-operative cytokine levels to IRI [11, 30, 31], there are limited data regarding the association of peri-reperfusion cytokine levels to EAD, particularly peri-reperfusion [35, 36]. We hypothesize perturbed cytokine, protein, and hypoxia inducible factor-1 alpha (HIF-1α) levels correlate with EAD. The identification of these aberrations may be used as novel biomarkers to assist in the development of new diagnostic approaches and therapeutic interventions for the identification, prevention, and management of EAD.

## Methods

### Study aims

Our central hypothesis is peri-reperfusion hyperoxemia and immune dysregulation exacerbate IRI and increase the risk of EAD. Our first aim is to determine the association between peri-

reperfusion hyperoxemia and EAD after deceased donor OLT. We will examine peri-reperfusion median serum $PaO_2$ levels to determine the odds EAD develops in patients exposed to hyperoxemia compared to those not exposed. Our second aim is to evaluate whether peri-reperfusion perturbed cytokine, protein, and HIF-1α levels correlate with EAD after OLT. We will measure peri-reperfusion serum cytokine, protein, and HIF-1α levels before and 2 and 48 hours after reperfusion for comparison between those who develop EAD and those who do not using the same participants from aim one. We will perform a subset analysis to determine whether peri-reperfusion hyperoxemia, compared to non-hyperoxemia, alters their pattern of expression.

## Design

This study is a pilot prospective single-center observational cohort study to determine (1) the association between peri-reperfusion hyperoxemia and EAD and (2) whether peri-reperfusion perturbed cytokine, protein, and HIF-1α levels correlate with EAD after deceased donor OLT. An observational design facilitates: (1) the examination of $PaO_2$ within the confines of existing variability in anesthesiologist-administered peri-reperfusion $FiO_2$, (2) prevents randomizing subjects to a potential harm, and (3) allows for the inclusion of patients with intrapulmonary shunting, which is common in patients with liver failure due to hepatopulmonary syndrome. This study does not meet criteria for a clinical trial per the National Institutes of Health (NIH) guidelines because it is observational and does not prospectively assign participants to an intervention to evaluate the effects of that intervention. A general study timeline is outlined in "Fig 2."

## Setting, key personnel, and facilities

This study will be performed at Emory University Hospital (EUH), a large academic hospital affiliated with Emory University School of Medicine in Atlanta, GA, USA. The principal investigator (PI) is a liver transplant anesthesiologist at EUH that will participate in and be responsible for the overall conduct of the study. The study team will consist of the PI, PI's mentors, other liver transplant anesthesiologists and anesthesiology residents at our institution, laboratory technicians, and a study coordinator. All study team members will be certified to conduct research through the Collaborative Institutional Training Initiative (CITI) Program.

Biosample processing, immune assay batch analysis, and biosample long-term storage will occur in a laboratory at EUH. Immune assay kits will be purchased through Meso Scale Discovery, a division of Meso Scale Diagnostics LLC, Rockville, MD.

## Eligibility criteria

Inclusion criteria include age $\geq$ 18 years, end-stage liver disease, and donation after brain or circulatory death. Exclusion criteria include living donor donation, repeat OLT within a week of transplantation, multiple organ transplantation, and pregnancy or lactating patients. Eligibility criteria are summarized in "Table 2." Prisoners do not meet criteria for OLT at our institution and are not included in this study. Donors after circulatory death who undergo NRP, a form of mechanical circulatory support similar to extracorporeal membrane oxygenation which restores oxygenated blood flow to the allograft after cardiac arrest and attenuates ischemic injury before recovery, will be included in the study. Patients who receive an allograft maintained on machine perfusion during storage, which re-exposes the allograft to oxygen before implantation, will also be included in the study. To limit the early re-exposure to oxygen as a confounding variable, these patients will be analyzed separately from those who receive an allograft maintained in static cold storage, the conventional method of allograft storage.

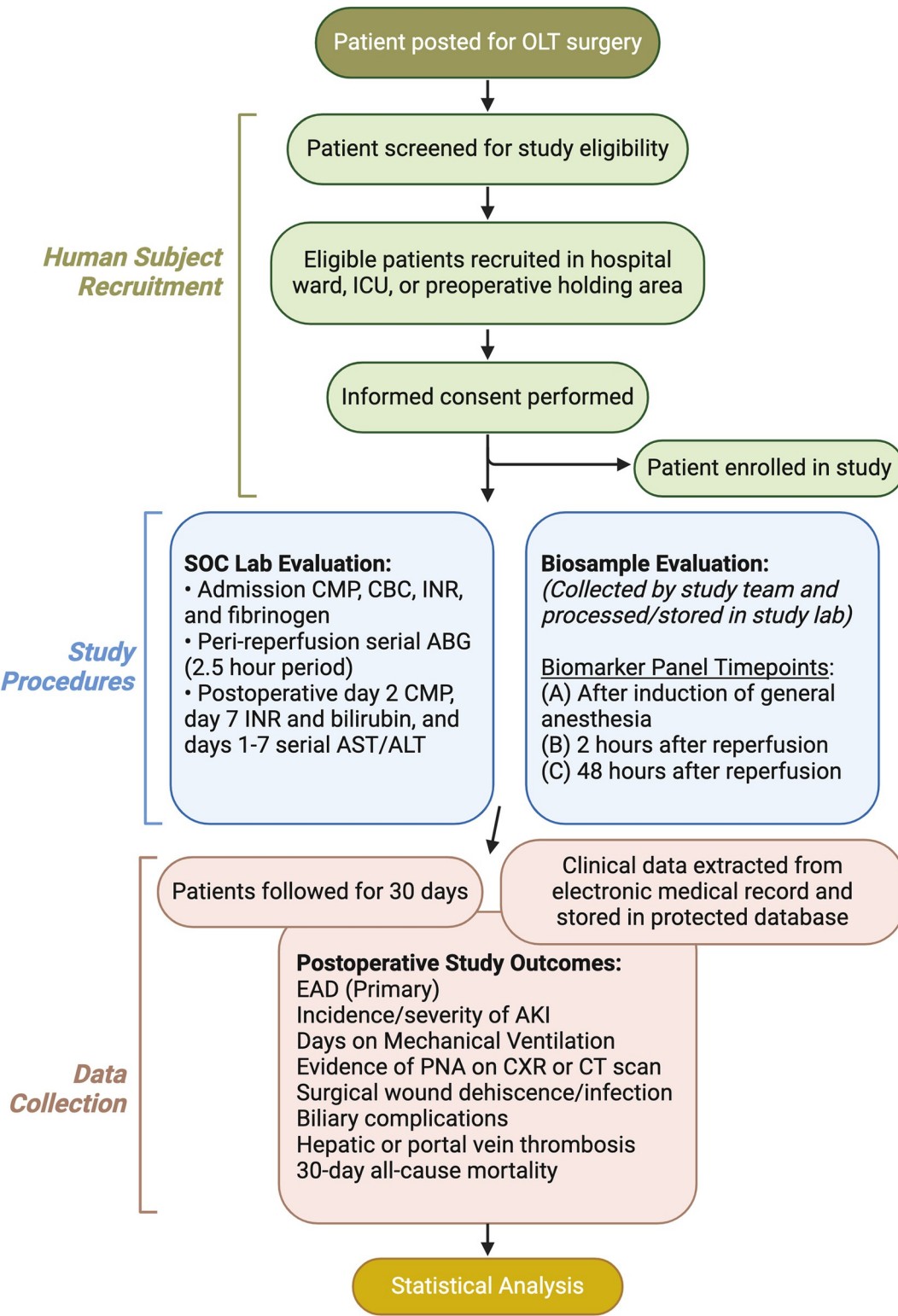

**Fig 2. General study timeline.** OLT indicates orthotopic liver transplantation; ICU, intensive care unit; SOC, standard of care; CBC, complete blood count; CMP, comprehensive metabolic panel; INR, international normalized ratio; ABG, arterial blood gas; AST, aspartate aminotransferase; ALT, alanine transaminase; EAD, early allograft dysfunction; AKI, acute kidney injury; PNA, pneumonia; CXR, chest x-ray; CT scan, computed tomography scan. Figure created with Biorender.com.

**Table 2. Eligibility criteria.**

| Inclusion Criteria | Exclusion Criteria |
| --- | --- |
| Age $\geq$ 18 years | Living donor donation |
| End-stage liver disease | Repeat OLT within a week of transplantation |
| Donation after brain death | Multiple organ transplantation |
| Donation after circulatory death | Pregnancy or lactating patients |

## Outcomes

The widely accepted Olthoff et al. definition of EAD is the primary outcome, specifically the presence of one or more of the following serum laboratory values: total bilirubin level $\geq$ 10 mg/dL on day 7, international normalized ratio (INR) level $\geq$ 1.6 on day 7, and peak aspartate transaminase (AST) or alanine transaminase (ALT) level > 2,000 IU/L within the first 7 post-operative days [1]. Secondary outcomes include: (1) the incidence and severity of postoperative acute kidney injury (AKI) as determined by the Kidney Disease Improving Global Outcomes (KDIGO) criteria, (2) number of postoperative days on mechanical ventilation, (3) evidence of postoperative pneumonia on chest x-ray or computed topography scan within the first 30 days postoperatively, (4) surgical wound dehiscence or infection within the first 30 days postoperatively, (5) anastomotic or non-anastomotic biliary strictures, biliary leak, bile duct stones or sludge, or ischemic cholangiopathy within the first 30 days postoperatively, (6) postoperative hepatic artery or portal vein thrombosis within the first 30 days postoperatively, and (7) 30-day all-cause mortality. Postoperative outcomes are summarized in "Table 3."

## Study measures

Peri-reperfusion oxygen exposure will be assessed by the median $PaO_2$ level 30 minutes before through 2 hours after reperfusion, reflecting the initial phase of reperfusion injury. $PaO_2$ will be examined in three ways: (1) as a continuous variable, (2) as a categorical variable dichotomized by hyperoxemia vs. non-hyperoxemia ($\geq$ 200 mmHg vs. < 200 mmHg), and (3) with quartile measurements.

Oxygen exposure will be examined by serum $PaO_2$ instead of anesthesiologist-administered $FiO_2$ for the following reasons: (1) $PaO_2$ better depicts the oxygen level to which the allograft is exposed due to the high incidence of intrapulmonary shunting in patients with liver failure and (2) it allows us to observe actual oxygen exposure as opposed to anesthesiologist intervention. Though an anesthesiologist may give peri-reperfusion $FiO_2$ 100%, allografts of recipients with a high burden of intrapulmonary shunting will be exposed to less oxygen due to the mixture of hypoxic and hyperoxic blood. Generally, however, if there is an absence or small

**Table 3. Primary and secondary outcomes.**

| Primary Outcome | Secondary Outcomes |
| --- | --- |
| EAD | Incidence and severity of postoperative AKI |
| | Number of postoperative days on mechanical ventilation |
| | Evidence of postoperative pneumonia on chest x-ray or computed topography scan within first 30 days postoperatively |
| | Surgical wound dehiscence/infection within first 30 days postoperatively |
| | Biliary complications within first 30 days postoperatively |
| | Postoperative hepatic artery or portal vein thrombosis within first 30 days postoperatively |
| | 30-day all-cause mortality |

**Table 4. Serum biomarker panel with rationale.**

| Biomarker Panel | Rationale |
| --- | --- |
| IL-1β, TNFα | Proinflammatory, promote neutrophil infiltration |
| IL-2R, IP-10, MIG | Proinflammatory, involved in T cell immunity |
| IL-8, MCP-1, RANTES | Proinflammatory, involved in NF-κB pathway |
| PTX3 | Activates complement, chemoattractant |
| IFN-γ | Proinflammatory, suppresses PTX3 |
| IL-6, HO-1 | Anti-inflammatory, reduces generation of ROS |
| HIF-1α | Hypoxia-signaling, regulates cytokine expression |

[11, 30, 34, 37–40] IL indicates interleukin; TNFa, tumor necrosis factor alpha; IP-10, interferon gamma-induced protein 10; MIG, monokine induced by gamma interferon; MCP-1, monocyte chemoattractant protein-1; RANTES, regulated on activation, normal T cell expressed and secreted; PTX3, pentraxin-related protein 3; IFNg, interferon-gamma; HO-1, heme oxygenase-1; HIF-1a, hypoxia-inducible factor-1alpha.

burden of intrapulmonary shunts, the higher the $FiO_2$ administered, the higher the $PaO_2$. In non-ventilated patients without supplemental oxygen, normal $PaO_2$ is 75–100 mmHg and hyperoxemia is $PaO_2 > 100$ mmHg. All OLT recipients are ventilated and receive some degree of supplemental oxygen. Because there is no standard definition of hyperoxemia in ventilated patients, we chose a threshold of 200 mmHg based on prior data in animal models and LDLT [23, 24]. Even with lower $FiO_2$ levels and some degree of intrapulmonary shunting, many OLT recipients have a $PaO_2 > 100$ mm Hg. Furthermore, a threshold of 200 mmHg would permit meaningful interventions by the anesthesiologist if identified as a risk factor for EAD.

Potential biomarkers of EAD, including serum cytokines, inflammatory proteins, and the transcription factor HIF-1α, were selected for evaluation based on biologic rationale and prior data in IRI "Table 4" [11, 30, 34, 37–40]. The peripheral serum level of these biomarkers will be compared between those with and without EAD.

## Sample size calculation

A total enrollment of 40 participants is planned. A post-hoc power analysis was performed based on exploratory data from Friedman et al. using the estimated mean and standard deviation (SD) of IL-8 on day 1 in the EAD and non-EAD groups to calculate effect size [34]. Since only median values and IQR were available, mean and SD of IL-8 for each group were estimated as median and IQR/1.35, respectively. A two tail post-hoc power analysis using an effect size of 0.717, alpha of 0.1, EAD sample size of 29, and non-EAD sample size of 44 achieved 84% power. To achieve power of 70% (A priori) with the same parameters, a total sample size of 40 is needed with at least 20 in each group. Given the pilot design of this study, a total sample size of 40 patients will be enrolled to generate preliminary data. This is feasible as our institution performs an average of 118 adult deceased donor OLT cases per year (Organ Procurement and Transplantation Network [OPTN], 2012–2021 data). Efforts will be made to enroll all eligible participants. There is no pre-determined percentage of subjects according to sex, gender, race, or ethnicity.

## Human subject recruitment

The Emory Transplant Center at EUH will announce an OLT case via encrypted email and/or private phone call to the PI, typically 24–48 hours before transplantation. The PI will disseminate this information to study team members involved in screening patient eligibility and obtaining informed consent. Patients will be screened for study eligibility via manual

electronic medical record chart review. Eligible patients will then be recruited for study enrollment in their respective location, either the hospital ward, intensive care unit (ICU), or pre-operative holding area. The study team will obtain informed consent by reviewing the study purpose, protocol, possible risks and benefits, right to refuse to partake, and right to withdraw in a confidential space. The patient or their legal representative will be given time to ask questions and have their questions answered in a satisfactory manner. Eligible subjects will be ensured their medical care does not change whether they participate in the study or not. If the subject agrees to participate, he or she will provide written informed consent per accordance with the ethical principles from the Declaration of Helsinki.

## Procedures

This study was approved by the Emory University institutional review board (IRB), protocol number 5728. Study enrollment began on August 1, 2023. At the time of publication, this study has enrolled thirty-four patients and no data analysis has been performed. A serum lab protocol capturing $PaO_2$ on serial arterial blood gas (ABG) analysis, other standard of care (SOC) tests, and cytokine, protein, and HIF-1$\alpha$ (biomarker) levels via serum biosample collection was implemented "Table 5." Peri-reperfusion serial ABG will be obtained ~30 minutes before through ~2 hours after reperfusion within the SOC.

The study team will collect peripheral serum biosamples at planned intervals to test the biomarker panel. Biosamples will be obtained via an arterial line, placed preoperatively as a SOC procedure. All intraoperative serum biosamples will be obtained by the anesthesiologist. All postoperative serum biosamples will be drawn by the ICU or inpatient floor nurse. The other tests are SOC and will be obtained and analyzed in their usual fashion. Extra labs will be obtained at the discretion of the anesthesiologist or intensivist within the SOC.

At each time point outlined for biosample collection, about 12 mL of blood will be collected into 2 edetic acid (EDTA) tubes (about 6 mL in each tube). Each subject will have 3 blood draws, about 36 mL of blood collected in total. Within 30 minutes of collection, serum biosamples will be processed via centrifugation at 4°C at 2,000g for 20 minutes to separate the plasma from the blood cells. The upper layer of plasma will be transferred via pipet to 1.5 mL cryovials. Upon centrifuging, every 4 mL of blood makes about 2 mL of centrifuged plasma. Thus, 12 mL of blood should make about 6 mL of centrifuged plasma, which will then be aliquoted to four 1.5 mL cryovials. A summary of serum biosample collection and processing is in "Table 6." All tubes and cryovials will be labeled with the date and time of collection, protocol

**Table 5. Subject serum lab evaluation timeline.**

| Time Point | Serum Laboratory Values |
|---|---|
| Admission | CMP, CBC, INR, Fibrinogen (SOC) |
| After anesthesia induction | Biomarker Panel |
| Peri-reperfusion | Serial ABG (SOC) |
| ~2 hours after reperfusion | Biomarker Panel |
| ~48 hours after reperfusion | Biomarker Panel |
| Postoperative day 2 | CMP (SOC) |
| Postoperative day 7 | INR, Bilirubin, AST/ALT (SOC) |
| Postoperative days 1–7 | Serial AST/ALT (SOC) |

CMP indicates complete metabolic panel; CBC, complete blood count; INR, international normalized ratio; SOC, standard of care; ABG, arterial blood gas; AST, aspartate aminotransferase; ALT, alanine transaminase. *The study team will only collect biosamples for the biomarker panel.*

**Table 6. Summary of serum biosample collection and processing.**

| Timepoint | Timepoint Letter | Patient Location | Source | Biosample and Tubes | Samples Stored |
|---|---|---|---|---|---|
| After Induction of General Anesthesia | A | OR | Peripheral Blood | ~12 mL total (~6 mL in 2 purple top EDTA tubes) | 4 cryovials of EDTA plasma (1.5 mL each) |
| 2 Hours after Reperfusion | B | OR | Peripheral Blood | ~12 mL total (~6 mL in 2 purple top EDTA tubes) | 4 cryovials of EDTA plasma (1.5 mL each) |
| 48 Hours after Reperfusion | C | ICU or Inpatient Floor | Peripheral Blood | ~12 mL total (~6 mL in 2 purple top EDTA tubes) | 4 cryovials of EDTA plasma (1.5 mL each) |

EDTA indicates edetic acid.

number, study identification number, and timepoint letter. Biosamples will be stored in a -80° freezer in the laboratory for later batch analysis. Standard kits to test these biomarkers, including Meso Scale Discovery Assay kits and multiplex ELISA, will be performed in duplicate in a laboratory per standardized manufacturer guidelines.

## Data collection and management

Participants will be followed for up to 30 days with weekly capture of clinical data extracted from the electronic medical record. Key clinical, laboratory, and radiology results will be accessed within the electronic medical record. Collected data will include relevant exposures, covariates, and outcomes. Covariates include known risk factors for EAD, peri-reperfusion anesthesiologist-administered $FiO_2$, peri-operative immunosuppressive agents, and the use of NRP and/or allograft machine perfusion. Biosamples will be labeled with the date, time, study protocol number, study identification number, and timepoint. Data will be stored in an encrypted, password-protected, de-identified database for the duration of the study and as per institutional and journal publication policies. Biosamples will be banked and managed per institutional policy. Only the PI will have access to a restricted, password-protected file which lists study identification numbers to corresponding participant identifiers and biosample information. Only the study personnel will have access to the long-term laboratory freezer storing the biosamples. The study team will only have access to the de-identified database. Data analysis will be performed with the de-identified dataset only.

## Statistical analysis

Patient and transplant characteristics will be reported descriptively (median and range or proportions). Patients will be grouped by the development of EAD or non-EAD. A type I error of 0.05 will be used as the criterion for statistical significance. Analyses will be performed using the R project for statistical computing.

For aim one, the proportion of patients exposed to hyperoxemia in each group, EAD or non-EAD, will be compared. Values will be presented as the median with interquartile range (IQR) or mean with standard deviation (SD) for continuous variables as appropriate and as the total number (percentage) for categorical variables. Comparisons between groups will be made using the Student's t test or Mann-Whitney U test for continuous variables and chi-square test or Fisher's exact test for categorical variables as appropriate. A multivariate logistic regression analysis will be performed with other EAD risk factors (e.g., donor type and age, ischemia time, and MELD score).

For aim two, cytokine, protein, and HIF-1α levels will be compared between the two groups, EAD and non-EAD, using analysis of variance (ANOVA) with Tukey's adjustment for *p*-values at each time point and over multiple time points. For each biomarker, patients will be categorized into quartiles based on measured concentration, and the odds ratio for EAD will be compared among quartiles using a logistic regression analysis. The lowest quartile for proinflammatory cytokines (or highest quartile for anti-inflammatory cytokines) will be used as the reference group to calculate adjusted and unadjusted odds ratios for EAD. We will develop a logistic regression model of EAD as a function of inflammation, using the area under the receiver operating characteristic (ROC) curve and the Hosmer-Lemeshow good-ness-of-fit test to assess model discrimination and calibration, respectively. We will perform a subset analysis to determine whether peri-reperfusion hyperoxemia, compared to non-hyper-oxemia, alters their pattern of expression.

NRP during DCD and machine perfusion of the allograft during storage, which limit ische-mia and allow early re-exposure to oxygen, are associated with a reduced incidence of EAD and improved postoperative outcomes [41–45]. Consequently, patients who receive an allo-graft that underwent NRP or stored on a machine perfusion pump will be analyzed separately from those who receive an allograft stored via static cold storage, the conventional and most widely used method of allograft storage worldwide. At our institution, only a small portion of allografts undergo NRP and normothermic machine perfusion is predominately used for DCD allografts that did not undergo NRP. Thus, DCD allografts at our institution either undergo NRP or are placed on a normothermic machine perfusion pump, but not both.

## Risks to participants

There are minimal risks to participants, namely breach of confidentiality/privacy and minimal blood loss obtained for the serum biosamples for the biomarker panel at each time point. Par-ticipants will have an arterial line placed as a SOC and biosamples will be obtained from this existing line. Thus, no additional pain, bruising, or bleeding due to an additional needlestick will occur when obtaining the biosamples. Blood is drawn frequently from arterial lines for clinical care and thus this study will not increase the risks associated with SOC testing. The study's risks to participants are both minimal and consistent across the entirety of the planned study population.

## Adequacy of protection against risks

To lessen the probability and/or magnitude of risk of breach of confidentiality, the following procedures will be instituted: (1) in-person informed consent will be obtained in a private space without coercion, allowing adequate time for questions, (2) only IRB-approved consent forms will be used, (3) signed consent forms will be stored in a locked file cabinet separate from the study database with a restricted file linking participant identifiers with their corre-sponding study identification number, (4) clinical care will not be altered whether patients decide to participate or not, (5) participants can withdraw from the study at any time, (6) the study database will be de-identified, restricted, and password-protected, and (7) only the mini-mal amount of blood required will be collected. This study will abide by the institutions' data security policy per our IRB.

## Vulnerability of study population

The study population includes primary, adult deceased donor OLT recipients at our institu-tion. Children, pregnant women, lactating women, and prisoners will not be included in this study. Patients meeting inclusion criteria may be members of particular special populations,

including institutional employees, adults unable to consent (e.g., inability to communicate due to critical illness or altered mental status), cognitively impaired individuals, those with impaired decision-making capacity, and/or individuals not able to understand English. Members of these special groups are not a specific focus of this study. Nonetheless, to optimize safety for these potential participants, the following protocol will be enforced: (1) employee status will not be collected as data, (2) only a legal surrogate will be able to give informed consent for study participation for those patients unable to consent, with cognitive impairment, or impaired decision-making capacity, (3) for those patients not able to clearly understand English, they will only be considered for enrollment if the informed consent process occurs in their primary language through an official interpreter, (4) all data will be de-identified and stored in a restricted database prior to analysis, and (5) participants will be withdrawn if they appear to be unduly distressed.

This research does not involve individuals who are vulnerable to coercion or undue influence. All patients who undergo liver transplantation at our institution undergo a rigorous, standardized, and formal review process for liver transplant candidacy prior to surgery. Vulnerable populations are not being targeted for enrollment in this study.

Gender, race, and ethnicity are plausible covariates in the research questions of this study and are planned for inclusion in data collection and analysis. Women and members of minority, non-white ethnic groups comprise approximately 35% and 30%, respectively, of all adult OLT cases performed annually [46, 47]. Prior data suggest women and racial or ethnic minority groups experience disparities with respect to access and outcomes in the perioperative OLT setting [46, 47]. Study team educational meetings and debriefings will occur to increase awareness of these disparities and promote equal access to study participation.

This study will employ JAMA definitions for "race" and/or "ethnicity," in which (1) the source of classification is described (i.e., white, black, etc. as reported by the Emory Transplant Center and electronic medical record based on the US Office of Management and Budget's Revisions to the Standards for the Classification of Federal Data on Race and Ethnicity), (2) reasons for assessment delineated (given known gender and ethnic disparities and the mandate by the NIH to report such data consistent with the Inclusion of Women, Minorities, and Children policy), and (3) race/ethnicity of the study population is reported in the results section of any manuscript. Racial and ethnic classifications of subjects will be used for descriptive statistics.

## Potential benefits of the research

Due to its observational design, participants will have no direct benefit through participation in this study. However, the information obtained from this study may serve as generalizable knowledge that advances the care of deceased donor OLT recipients in the future.

## Oversight and monitoring

The study steering committee will include the PI, PI's mentors, and study coordinator. The PI and steering committee will monitor recruitment, enrollment, data collection and management, and oversee results reporting and publication. The committee will meet on a regular basis to oversee the study and ensure successful completion.

Given its observational design, this study is not deemed a clinical trial per current NIH guidelines and thus the appointment of a formal data monitoring committee is not required. Nonetheless, all potential adverse events will be recorded by the study team and reported to our IRB. The consent form describes how participants can report any concerns regarding participation in the study.

If an enrolled participant decides to withdraw from the study, their clinical data and bio-samples will be removed from the study. This option is detailed in the consent. We do not anticipate circumstances under which participants will be withdrawn from the study without their consent.

## Discussion

### Association of hyperoxemia and EAD

We anticipate (1) increasing peri-reperfusion $PaO_2$ is associated with EAD, (2) the odds of EAD increase when exposed to peri-reperfusion hyperoxemia compared to non-hyperoxemia, (3) $PaO_2$ quartiles may have a differential effect on EAD, and (4) certain sub-populations may have higher odds of developing EAD when exposed to hyperoxemia. Such findings would lend evidence to an understanding of EAD as an inflammatory condition exacerbated by the re-introduction of excess oxygen. If validated, anesthesiologists can titrate $FiO_2$ to a goal $PaO_2$ to mitigate the impact of EAD and improve postoperative outcomes after OLT.

Nonetheless, our study protocol has a few limitations. As a pilot study, our protocol may be underpowered to detect a change. Alternatively, our study may not capture the key timeframe linked to the risk of EAD after oxygen re-exposure. It is possible oxygen exposure beyond the first 2.5 peri-reperfusion hours may be important for IRI and the development of EAD. Additionally, it is possible peri-reperfusion hyperoxemia confers increased risk only in a particular subgroup of patients undergoing OLT, e.g., recipients with metabolic dysfunction-associated steatohepatitis (MASH). While this pilot study will be insufficient to answer these questions, trends in data may serve to alter a subsequent larger study and address any of these findings.

### Association of immune dysregulation and EAD

We anticipate (1) distinct, perturbed cytokine, protein, and HIF-1α levels are associated with the development of EAD, (2) recipients who develop EAD have greater proinflammatory activity and/or less anti-inflammatory activity than those without EAD, and (3) peri-reperfusion hyperoxemia alters the pattern of expression of cytokine, protein, and HIF-1α levels. Such findings would lend evidence to an understanding of EAD as an inflammatory condition characterized by pathologic dysregulation of innate immune defenses with the re-introduction of oxygen. If validated, future studies could investigate the peri-reperfusion administration of immunomodulatory agents against specific perturbed cytokines, proteins, and/or HIF-1α to mitigate the impact of EAD and improve postoperative outcomes after OLT.

This aim also has limitations. As a pilot study, our protocol may be underpowered to detect a change. It is also possible there is no correlation between the tested peri-reperfusion bio-markers levels and EAD because the key biomarkers are not in the selected panel or biomarker levels are altered outside our studied timeframe. Given the inclusion of biomarkers linked to IRI in our panel, we plan to analyze the biomarker panel in the first 13 patients and if there is no signal, we plan to include a later time point of biosample collection.

Biomarker levels may be affected by the intraoperative and postoperative administration of immunosuppressive agents. At our institution, a dose of methylprednisolone is given intraoperatively during the neohepatic phase after hemostasis is achieved. Thereafter, a short course of methylprednisolone is initiated on the first postoperative night. Postoperatively, standing mycophenolate mofetil is started the evening of day zero and tacrolimus on day one or two depending on the patient's baseline renal function. While these doses are relatively constant, another agent, basiliximab, is administered intraoperatively after methylprednisolone for patients with a creatinine clearance of $< 60$ mL/min to delay initiation of reduced-dose

tacrolimus in an effort to reduce kidney injury [48]. The specific immunosuppressive agents and timing of administration will be recorded for each patient.

Our timepoints for serum biosample collection were chosen to minimize variability in immunosuppressant exposure. The first serum biosample is obtained prior to the administration of any immunosuppressive agents. The second serum biosample, obtained 2 hours after reperfusion, is most likely drawn after the administration of intraoperative methylprednisolone and possibly basiliximab depending on the degree of bleeding, resuscitation required, and baseline renal function. The third serum biosample, obtained 48 hours after reperfusion, will be drawn after methylprednisolone and mycophenolate mofetil exposure, and tacrolimus for most patients as well. We will explore if the administration and timing of immunosuppressive agents affects biomarker levels and may modify the patients captured to minimize variability if there is a signal.

Finally, our results may be impacted by large volume blood product resuscitation due to coagulopathy during OLT. Participant blood turnover with donated blood products may affect or "wash-out" our biomarker levels. This would most likely lead to low levels of all tested biomarkers at the second and third timepoints. If the first serum biosamples analyzed demonstrate this effect, we will consider altering the timepoints tested to focus on early events. Alternatively, if a specific biomarker showed statistically significant change over the course of the protocol despite high volume blood product resuscitation, it could provide unique insight into the immune dysregulation associated with EAD.

## Unique challenges assaying serum HIF-1α

Investigations of serum HIF-1 α have unique challenges because it is a transcription factor. Unlike extracellular cytokines, HIF is an intracellular heterodimeric transcription factor with one of three tightly regulated α subunits (HIF-1α, HIF-2α, and HIF-3α) that binds to an oxygen-dependent β subunit. During normoxia, cytoplasmic HIF-1α α-subunits are degraded by factor inhibiting HIF (FIH) and prolyl hydroxylase domain (PHD) enzymes via a von Hippel-Lindau polyubiquitination, protease-dependent pathway [49]. Conversely, during hypoxia, FIH and PHD are inhibited and HIF-1α levels increase, partly due to protein stabilization [50]. HIF-1α subsequently undergoes translocation from the cytoplasm into the nucleus where it induces the transcription of genes involved in cellular metabolism, glycolysis, angiogenesis, cytoskeletal formation, and hormonal regulation [50]. HIF-1α signaling can quickly reverse with normoxia [50].

HIF-1α has largely been studied in tissue via protein quantification with Western blotting or immunohistochemistry [50]. Immunohistochemistry studies are advantageous because they can identify the cellular location of HIF-1α, which aids in the interpretation of the role of this transcription factor in the disease process. Conversely, the Meso Scale Discovery total HIF-1α assay that will be employed in our protocol uses whole blood lysate electrochemiluminescence technology and a sandwich immunoassay, similar to ELISA. This method examines circulating HIF-1α levels, precluding intracellular location, with the presumption intracellular HIF-1α is released extracellularly into the blood after endothelial injury and cell death. This should occur in our study as IRI causes hepatocellular damage, sinusoidal endothelial injury, and cell death.

The extracellular release kinetics of HIF-1α following endothelial injury have been assessed in a rat model [51]. In this study, HIF-1α levels in cell-culture medium and plasma were measured via ELISA and the results were validated using Western blotting and immunohistochemistry. HIF-1α levels reached maximum concentrations approximately 2 hours post injury *in vitro* and 2 days post injury *in vivo* [51]. Clinically, elevated circulating levels of HIF-1α are linked to cellular injury in other disease processes in renal, cardiovascular, and cancer research [52–57].

Despite these challenges, it is valuable to examine HIF-1α because it is widely recognized as the universal regulator of the cellular response to hypoxia and interacts with inflammatory cytokines, many of which we are studying. The protective effects of ethyl 3,4-dihyroxybenzoate, a PHD inhibitor and anti-oxidant, demonstrated mitochondrial protection and mitigated hepatocellular necrosis in a partial liver IRI mouse model [58]. Additionally, HIF-1α upregulates HO-1, a cytoprotective molecule in IRI, and TNFα, a proinflammatory cytokine central to IRI, and cross-communicates with the NF-κB pathway [50]. HIF-1 α is thus a potential therapeutic target during OLT as hypoxemia and ischemia are hallmarks of IRI.

## Conclusion

Our overarching goal is to decrease the risk and impact of EAD and improve postoperative outcomes after OLT. Characterizing the peri-reperfusion factors associated with EAD could enhance our understanding of this process, permitting identification of those at risk in real time and potentially leading to targeted anesthetic interventions to decrease tissue injury and improve postoperative outcomes and access to OLT.

## Acknowledgments

The authors would like to thank many staff for their contribution to this project: (1) the Georgia Clinical & Translational Science Alliance (CTSA) Clinical Research Center (GCRC) for its contribution to the collection and processing of serum biosamples, (2) the Emory Research Hemostasis & Coagulation Core Laboratory (ERHCCL) for its contribution to biosample storage and biomarker panel analysis, and (3) anesthesiology study personnel for their contribution to participant enrollment.

## Author Contributions

**Conceptualization:** Elizabeth A. Wilson.

**Funding acquisition:** Elizabeth A. Wilson.

**Investigation:** Elizabeth A. Wilson.

**Methodology:** Elizabeth A. Wilson, Anna Woodbury, Kirsten M. Williams, Craig M. Coopersmith.

**Project administration:** Elizabeth A. Wilson.

**Resources:** Anna Woodbury, Kirsten M. Williams, Craig M. Coopersmith.

**Supervision:** Anna Woodbury, Kirsten M. Williams, Craig M. Coopersmith.

**Validation:** Anna Woodbury, Kirsten M. Williams, Craig M. Coopersmith.

**Visualization:** Elizabeth A. Wilson.

**Writing – original draft:** Elizabeth A. Wilson.

**Writing – review & editing:** Anna Woodbury, Kirsten M. Williams, Craig M. Coopersmith.

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
