## [Decision Letter · Decision Letter 0]

1 Feb 2024

PONE-D-23-27754OXIDATIVE study: a pilot prospective observational cohort study protocol examining the influence of peri-reperfusion hyperoxemia and immune dysregulation on early allograft dysfunction after orthotopic liver transplantationPLOS ONE

Dear Dr. Wilson,

Thank you for submitting your manuscript to PLOS ONE. After careful consideration, we feel that it has merit but does not fully meet PLOS ONE’s publication criteria as it currently stands. Therefore, we invite you to submit a revised version of the manuscript that addresses the points raised during the review process. Please submit your revised manuscript by Mar 17 2024 11:59PM. If you will need more time than this to complete your revisions, please reply to this message or contact the journal office at plosone@plos.org. Please include the following items when submitting your revised manuscript:A rebuttal letter that responds to each point raised by the academic editor and reviewer(s). You should upload this letter as a separate file labeled 'Response to Reviewers'.A marked-up copy of your manuscript that highlights changes made to the original version. You should upload this as a separate file labeled 'Revised Manuscript with Track Changes'.An unmarked version of your revised paper without tracked changes. You should upload this as a separate file labeled 'Manuscript'.

We look forward to receiving your revised manuscript.

Kind regards,

Pavel Strnad

Academic Editor

PLOS ONE

Journal Requirements:

Additional Editor Comments:

As you can see, both reviewers appreciated the design and the aims of your study and agree that a publication of the study protocol makes sense. At the same time, they make a couple of suggestions how to further improve the clarity of the current manuscript. 

Reviewers' comments:

Reviewer's Responses to Questions

**Comments to the Author**

1. Does the manuscript provide a valid rationale for the proposed study, with clearly identified and justified research questions?

Reviewer #1: Yes

Reviewer #2: Yes

2. Is the protocol technically sound and planned in a manner that will lead to a meaningful outcome and allow testing the stated hypotheses?

Reviewer #1: Yes

Reviewer #2: Yes

3. Is the methodology feasible and described in sufficient detail to allow the work to be replicable?

Reviewer #1: Yes

Reviewer #2: Yes

4. Have the authors described where all data underlying the findings will be made available when the study is complete?

Reviewer #1: Yes

Reviewer #2: Yes

5. Is the manuscript presented in an intelligible fashion and written in standard English?

Reviewer #1: Yes

Reviewer #2: Yes

6. Review Comments to the Author

You may also provide optional suggestions and comments to authors that they might find helpful in planning their study.

Reviewer #1: Early allograft dysfunction after orthotopic liver transplantation is a common complication after liver transplantation with an increased mortality. The underlying mechanisms are multifactorial with ischemia reperfusion injury as one of the key drivers of the early allograft dysfunction. The study design described in the manuscript aims to understand the role peri-reperfusion hyperoxemia in this context and tries to identify new markers that correlate with early allograft dysfunction. The authors designed a single-center observational cohort study that is well thought out and feasible to reach the goal of the study.

This manuscript from Wilson et. al. is summarizing a conclusive, interesting study design that fits in the scope of PlosOne.

1. The authors may add an illustration of the study design to make it easier for readers to understand the concept of the study.

Reviewer #2: The present manuscript is a study protocol for the “OXIDATIVE” trial, a prospective observational cohort study in deceased-donor liver transplantation. The main hypothesis of the present trial is that peri-reperfusion O2 levels may correlate with IRI and postoperative outcomes. I read the present article with great interest and commend the authors for their interesting research question. I believe that this study may provide great value to the scientific community. I recommend the acceptance of this manuscript, pending the following revisions:

Major:

1) To me, it was a little difficult to understand the exact timeline of sampling and interventions/ analyses in the first half of the paper. While the endpoints are named, a summary of “what happens when” during the study course should be added to the introduction. The description of IRI may be shortened a little in the introduction to accommodate for this.

I believe it would be important to briefly clarify the following aspects of the study design:

1a) l. 156 f: “We will examine peri-reperfusion median serum PaO2 levels to determine the odds EAD develops in patients exposed to hyperoxemia compared to those not exposed.” Is this a retrospective finding after the analysis of the peri-reperfusion sample? Serum PaO2 will probably correlate to the FiO2, is this somehow standardized, e.g. by an institutional protocol?

1b) The manuscript would benefit from a schematic representation of the study, in the form of a figure with a timeline. This should include the main analytic milestones of sampling and the timepoints of clinical visits/ evaluations (e.g. first 7 days for EAD, 30 days for surgical complications?). While some of this information is given in Table 4, a concise summarizing Figure in the early part of the paper will improve overall understanding.

1c) A table or flow-chart regarding inclusion and exclusion criteria would facilitate reading. Similarly, the endpoints can be additionally presented in tabular form.

2) The investigators plan on including a quite heterogenous collective of ECD (which are already potentially heterogenous regarding donor age, steatosis, pre-donation elevated liver enzymes), DCD (livers which suffer a very specific hypoxic-metabolic injury in the donation process) and non-ECD donor organs. This will surely impact the postoperative clinical outcomes. Are the investigators planning to adjust the statistical analysis to this fact?

Minor:

From l. 91, “IRI as a critical barrier”: perhaps insert a brief information that there are already strategies to mitigate IRI through O2 restoration in the donor organ through machine perfusion (HOPE), and that the pre-implantation restoration of oxygen in the donor organ is a central tool for improving postoperative outcomes. There is an accumulating body of clinical evidence from RCTs regarding HOPE (e.g. Schlegel et al, 2023, PMID: 36681160; Czigany et al, 2021, PMID: 34334635). I believe this would strengthen the hypothesis that oxygen levels in the liver may impact IRI differently than systemic ones, and are critical for postoperative outcomes. MP is mentioned later, e.g. l. 355ff, could you comment how this is applied? End-ischemically? HOPE or normothermic?

Liver transplant recipients exhibit extremely variable degrees of morbidity, from clinically stable patients with HCC within Milan criteria, to patients with decompensated cirrhosis or ACLF. Can the authors comment on this and does this affect intraoperative O2 management? Will preoperative disease severity be considered when analyzing postoperative outcomes? Is this implicated in l. 318? Perhaps this should be explicitly mentioned as it constitutes a central difficulty in designing clinical trials in transplantation.

l. 315: follow-up of 4 weeks does not allow for a 30-day morbidity assessment. Is follow-up 30 days, then?

7. PLOS authors have the option to publish the peer review history of their article (what does this mean?). If published, this will include your full peer review and any attached files.

Reviewer #1: No

Reviewer #2: No

---

## [Author Response · Author response to Decision Letter 0]

12 Feb 2024

OXIDATIVE study response to reviewers

Changes in the revised manuscript with track changes are noted in red.

As requested, the formatting of the revised manuscript with track changes and manuscript has been updated per PLOS One style requirements. 

Reviewer's Responses to Questions

Comments to the Author

1. Does the manuscript provide a valid rationale for the proposed study, with clearly identified and justified research questions?

Reviewer #1: Yes

Reviewer #2: Yes

2. Is the protocol technically sound and planned in a manner that will lead to a meaningful outcome and allow testing the stated hypotheses?

Reviewer #1: Yes

Reviewer #2: Yes

3. Is the methodology feasible and described in sufficient detail to allow the work to be replicable?

Reviewer #1: Yes

Reviewer #2: Yes

4. Have the authors described where all data underlying the findings will be made available when the study is complete?

Reviewer #1: Yes

Reviewer #2: Yes

5. Is the manuscript presented in an intelligible fashion and written in standard English?

Reviewer #1: Yes

Reviewer #2: Yes

6. Review Comments to the Author

You may also provide optional suggestions and comments to authors that they might find helpful in planning their study.

Responses to reviewers in this document are noted in red and reflected in the revised manuscript with track changes in red.

Reviewer #1: Early allograft dysfunction after orthotopic liver transplantation is a common complication after liver transplantation with an increased mortality. The underlying mechanisms are multifactorial with ischemia reperfusion injury as one of the key drivers of the early allograft dysfunction. The study design described in the manuscript aims to understand the role peri-reperfusion hyperoxemia in this context and tries to identify new markers that correlate with early allograft dysfunction. The authors designed a single-center observational cohort study that is well thought out and feasible to reach the goal of the study.

This manuscript from Wilson et. al. is summarizing a conclusive, interesting study design that fits in the scope of PlosOne.

1. The authors may add an illustration of the study design to make it easier for readers to understand the concept of the study.

An illustration of the study design has been added to the revised manuscript with track changes. It is referred to as “Fig 2” in the “Design” subsection of the “Methods” section.

Reviewer #2: The present manuscript is a study protocol for the “OXIDATIVE” trial, a prospective observational cohort study in deceased-donor liver transplantation. The main hypothesis of the present trial is that peri-reperfusion O2 levels may correlate with IRI and postoperative outcomes. I read the present article with great interest and commend the authors for their interesting research question. I believe that this study may provide great value to the scientific community. I recommend the acceptance of this manuscript, pending the following revisions:

Major:

1) To me, it was a little difficult to understand the exact timeline of sampling and interventions/ analyses in the first half of the paper. While the endpoints are named, a summary of “what happens when” during the study course should be added to the introduction. The description of IRI may be shortened a little in the introduction to accommodate for this.

An illustration of the study design/timeline has been added to the revised manuscript with track changes. It is referred to as “Fig 2” in the “Design” subsection of the “Methods” section.

I believe it would be important to briefly clarify the following aspects of the study design:

1a) l. 156 f: “We will examine peri-reperfusion median serum PaO2 levels to determine the odds EAD develops in patients exposed to hyperoxemia compared to those not exposed.” Is this a retrospective finding after the analysis of the peri-reperfusion sample? Serum PaO2 will probably correlate to the FiO2, is this somehow standardized, e.g. by an institutional protocol?

PaO2 values are reported on serial arterial blood gases during the liver transplant surgery. The blood samples for arterial blood gases are not part of the study sample as they are standard of care labs performed during all liver transplant surgeries. The serum laboratory values (standard of care tests and biomarker panel) time points are outlined in “Table 5” of the manuscript. This can be found in the “Procedures” subsection of the “Methods” section. 

During data collection, the median PaO2 within 2.5 hours of reperfusion (as outlined in the manuscript) will be collected and this will be used in data analysis as the study measure. 

Generally, PaO2 correlates with FiO2. However, most of our liver patients have intrapulmonary shunts in the setting of hepatopulmonary syndrome. This can create a discrepancy between the FiO2 set by the anesthesiologist and the oxygen in the blood to which the allograft is exposed. We want to examine the oxygen level to which the allograft is exposed because ischemia-reperfusion injury occurs in the allograft. This is why we chose PaO2 as the study measure, not FiO2. Though we acknowledge, generally speaking, as the FiO2 increases, the PaO2 increases. This is why a target PaO2 could potentially be established and an anesthesiologist could titrate the FiO2 to get to that target PaO2. 

This is explained in the “Study measures” subsection of the “Methods” section: “Oxygen exposure will be examined by serum PaO2 instead of anesthesiologist-administered FiO2 for the following reasons: (1) PaO2 better depicts the oxygen level to which the allograft is exposed due to the high incidence of intrapulmonary shunting in patients with liver failure and (2) it allows us to observe actual oxygen exposure as opposed to anesthesiologist intervention. Though an anesthesiologist may give peri-reperfusion FiO2 100%, allografts of recipients with a high burden of intrapulmonary shunting will be exposed to less oxygen due to the mixture of hypoxic and hyperoxic blood. Generally, however, if there is an absence or small burden of intrapulmonary shunts, the higher the FiO2 administered, the higher the PaO2.”

There is no regional, national, international, or institutional protocol for FiO2 or PaO2. There is marked variability in practice, as mentioned in the manuscript. This is one reason why we are performing the study. 

This is explained in the “Effect of hyperoxemia on EAD” subsection of “Introduction” section: “There is no standard of care concerning the optimal peri-reperfusion oxygen level during OLT. Furthermore, the impact of peri-reperfusion excess oxygen exposure in deceased donor OLT recipients has not been examined. A 2022 expert panel of liver transplant clinicians recommended a restrictive oxygen strategy targeting a PaO2 of 70-120 mmHg during OLT, citing evidence from non-OLT surgeries such as abdominal surgery. [25] These abdominal surgery studies demonstrate an increased risk of mortality, cancer, atelectasis, and elevated cardiac injury markers with higher perioperative oxygen concentrations. [25-29] However, without evidence in deceased donor OLT, this recommendation has not been widely implemented. Many OLT anesthesiologists administer a high peri-reperfusion FiO2 to mitigate cellular hypoxia during malperfusion, while others administer a low FiO2 to attenuate IRI.”

The study blood samples are only for the serum biomarkers. These are obtained after the induction of general anesthesia, 2 hours after reperfusion, and 48 hours after reperfusion. These blood draws are not part of the standard of care and are solely for the purposes of this study. This can also be found in “Table 5” as described above. 

1b) The manuscript would benefit from a schematic representation of the study, in the form of a figure with a timeline. This should include the main analytic milestones of sampling and the timepoints of clinical visits/ evaluations (e.g. first 7 days for EAD, 30 days for surgical complications?). While some of this information is given in Table 4, a concise summarizing Figure in the early part of the paper will improve overall understanding.

An illustration of the study design/timeline has been added to the revised manuscript with track changes. It is referred to as “Fig 2” in the “Design” subsection of the “Methods” section.

1c) A table or flow-chart regarding inclusion and exclusion criteria would facilitate reading. Similarly, the endpoints can be additionally presented in tabular form.

A table of eligibility criteria, “Table 2,” was added to the manuscript. A table of all outcomes, “Table 3,” was added to the manuscript. The original primary and secondary outcomes paragraphs were condensed into one paragraph, “Outcomes.”

2) The investigators plan on including a quite heterogenous collective of ECD (which are already potentially heterogenous regarding donor age, steatosis, pre-donation elevated liver enzymes), DCD (livers which suffer a very specific hypoxic-metabolic injury in the donation process) and non-ECD donor organs. This will surely impact the postoperative clinical outcomes. Are the investigators planning to adjust the statistical analysis to this fact?

ECD is a heterogenous category of donors. In fact, there is no regional, national, or international definition of what an ECD donor is. Institutions, regions, and countries define ECD differently. Thus, ECD is not standardized and is hard to track. It is more of a concept of presumably “higher risk” donors. The ability to determine allograft steatosis is also variable. Allograft biopsies are often performed per surgeon preference without any institutional, regional, national, or international standardization. DCD donors are known as “high risk” donors, in particular, though this may be changing with the advent of normothermic regional perfusion and machine perfusion, which is outside the scope of this manuscript. Donor type will be included in the multivariate analysis, in addition to other known risk factors for EAD (e.g., donor age, ischemia time, and recipient MELD score). 

To clarify this further, the following sentence was adjusted in the “Statistical analysis” subsection of the “Methods” section: A multivariate logistic regression analysis will be performed with other EAD risk factors (e.g., donor type and age, ischemia time, and MELD score). 

Minor:

From l. 91, “IRI as a critical barrier”: perhaps insert a brief information that there are already strategies to mitigate IRI through O2 restoration in the donor organ through machine perfusion (HOPE), and that the pre-implantation restoration of oxygen in the donor organ is a central tool for improving postoperative outcomes. There is an accumulating body of clinical evidence from RCTs regarding HOPE (e.g. Schlegel et al, 2023, PMID: 36681160; Czigany et al, 2021, PMID: 34334635). I believe this would strengthen the hypothesis that oxygen levels in the liver may impact IRI differently than systemic ones, and are critical for postoperative outcomes. MP is mentioned later, e.g. l. 355ff, could you comment how this is applied? End-ischemically? HOPE or normothermic?

A brief sentence about normothermic regional perfusion and machine perfusion have been added to the revised manuscript in the “IRI as a critical barrier” subsection of the “Introduction” section:

“To mitigate the impact of IRI and EAD, normothermic regional perfusion (NRP) and hypothermic and normothermic machine perfusion have been implemented to restore allograft oxygenation pre-implantation. This new technology, however, has limited global accessibility and thus static cold storage of the allograft on ice remains the conventional and most widely used method of allograft storage worldwide.” 

To be more comprehensive, we also included DCD donors who undergo normothermic regional perfusion and this is reflected in the revised manuscript in the “Eligibility criteria” subsection of the “Methods” section:

“Donors after circulatory death who undergo normothermic regional perfusion, a form of mechanical circulatory support similar to extracorporeal membrane oxygenation which restores oxygenated blood flow to the allograft after cardiac arrest and attenuates ischemic injury before recovery, will be included in the study.”

Those patients whose donors underwent normothermic regional perfusion, which is a small number at our institution and in the United States, will be analyzed separately, similar to those who received machine perfusion. This is exampled in the statistical section of the manuscript.

Machine perfusion, hypothermic and normothermic, has reduced the incidence of EAD. However, it is expensive and thus only certain areas of the world have access to it. In the United States, only normothermic machine perfusion has been approved by the FDA. The conventional and most widely used form of allograft storage is still static cold storage. For this study, we really want to look at static cold storage to eliminate normothermic regional perfusion and machine perfusion as covariables. However, we decided to include these cases to collect preliminary data for a future study. To focus the manuscript, we did not go into the details of normothermic regional perfusion or machine perfusion. There is a growing body of evidence supporting the use of these techniques (as mentioned in the manuscript) but this is not the primary objective of our study and with limited space, we decided to keep the manuscript more focused on our aims. 

This is reflected in the following revised sentence in the “Statistical analysis” subsection of the “Methods” section:

“NRP during DCD and machine perfusion of the allograft during storage, which limit ischemia and allow early re-exposure to oxygen, are associated with a reduced incidence of EAD and improved postoperative outcomes. [41-45] Consequently, patients who receive an allograft that underwent NRP or stored on a machine perfusion pump will be analyzed separately from those who receive an allograft stored via static cold storage, the conventional and most widely used method of allograft storage worldwide. At our institution, only a small portion of allografts undergo NRP and normothermic machine perfusion is predominately used for DCD allografts that did not undergo NRP. Thus, DCD allografts at our institution either undergo NRP or are placed on a normothermic machine perfusion pump, but not both.” 

*****Liver transplant recipients exhibit extremely variable degrees of morbidity, from clinically stable patients with HCC within Milan criteria, to patients with decompensated cirrhosis or ACLF. Can the authors comment on this and does this affect intraoperative O2 management? Will preoperative disease severity be considered when analyzing postoperative outcomes? Is this implicated in l. 318? Perhaps this should be explicitly mentioned as it constitutes a central difficulty in designing clinical trials in transplantation.

Intraoperative FiO2 management is determined by the anesthesiologist performing the case. There is no standard of care (as mentioned in the manuscript). An anesthesiologist may use individual judgement with different patients per their comorbidities, but is completely up to that anesthesiologist. There is marked variability in management among anesthesiologists. This is one reason why we want to perform this study as perhaps there should be an evidence-based standard of care in which FiO2 can be titrated by the anesthesiologist to a goal PaO2. This concept is described in the manuscript. 

Yes, preoperative disease severity will be considered when analyzing our outcomes. As mentioned, the multivariate analysis will include known risk factors for EAD, such as MELD score. This is stated in the manuscript. We are also collecting donor and recipient characteristics and they will be a part of the descriptive statistics. 

l. 315: follow-up of 4 weeks does not allow for a 30-day morbidity assessment. Is follow-up 30 days, then?

Patient’s charts will be followed for 30 days, not 4 weeks. This has been changed in the revised manuscript in the “Data collection and management” subsection of the “Methods” section:

“Participants will be followed for up to 30 days with weekly capture of clinical data extracted from the electronic medical record.”

---

## [Decision Letter · Decision Letter 1]

14 Mar 2024

OXIDATIVE study: a pilot prospective observational cohort study protocol examining the influence of peri-reperfusion hyperoxemia and immune dysregulation on early allograft dysfunction after orthotopic liver transplantation

PONE-D-23-27754R1

Dear Dr. Wilson,

We’re pleased to inform you that your manuscript has been judged scientifically suitable for publication and will be formally accepted for publication once it meets all outstanding technical requirements.

Kind regards,

Pavel Strnad

Academic Editor

PLOS ONE

Additional Editor Comments (optional): Thank you for this nice work!

Reviewers' comments:

Reviewer's Responses to Questions

**Comments to the Author**

1. Does the manuscript provide a valid rationale for the proposed study, with clearly identified and justified research questions?

Reviewer #1: Yes

Reviewer #2: Yes

2. Is the protocol technically sound and planned in a manner that will lead to a meaningful outcome and allow testing the stated hypotheses?

Reviewer #1: Yes

Reviewer #2: Yes

3. Is the methodology feasible and described in sufficient detail to allow the work to be replicable?

Reviewer #1: Yes

Reviewer #2: Yes

4. Have the authors described where all data underlying the findings will be made available when the study is complete?

Reviewer #1: Yes

Reviewer #2: Yes

5. Is the manuscript presented in an intelligible fashion and written in standard English?

Reviewer #1: Yes

Reviewer #2: Yes

6. Review Comments to the Author

You may also provide optional suggestions and comments to authors that they might find helpful in planning their study.

Reviewer #1: The authors addressed my comment and added an illustration of the study design to make it easier for readers to understand the concept of the study. I recommend the acceptance of the manuscript.

Reviewer #2: The present manuscript is now suitable for acceptance. The authors have included most suggestions into their revised version.

7. PLOS authors have the option to publish the peer review history of their article (what does this mean?). If published, this will include your full peer review and any attached files.

Reviewer #1: No

Reviewer #2: No

---

## [Editor Report · Acceptance letter]

18 Mar 2024

PONE-D-23-27754R1 

PLOS ONE

Dear Dr. Wilson, 

I'm pleased to inform you that your manuscript has been deemed suitable for publication in PLOS ONE. Congratulations! Your manuscript is now being handed over to our production team.

Kind regards, 

on behalf of

Dr. Pavel Strnad 

Academic Editor

PLOS ONE